# Improvement of Peptidyl Copper Complexes Mimicking Catalase: A Subtle Balance between Thermodynamic Stability and Resistance towards H_2_O_2_ Degradation

**DOI:** 10.3390/molecules27175476

**Published:** 2022-08-26

**Authors:** Yaqine Ben Hadj Hammouda, Koudedja Coulibaly, Alimatou Bathily, Magdalene Teoh Sook Han, Clotilde Policar, Nicolas Delsuc

**Affiliations:** 1Laboratoire des Biomolécules, LBM, Département de Chimie, Ecole Normale Supérieure, PSL University, Sorbonne Université, CNRS, 75005 Paris, France; 2Département de Chimie, Ecole Normale Supérieure, PSL University, 75005 Paris, France

**Keywords:** catalase mimic, di-copper(II) complexes, H_2_O_2_ dismutation, metal binding peptide, reactive oxygen species

## Abstract

Catalase mimics are low molecular weight metal complexes that reproduce the activity of catalase, an antioxidant metalloprotein that participates in the cellular regulation of H_2_O_2_ concentration by catalyzing its dismutation. H_2_O_2_ is a reactive oxygen species that is vital for the normal functioning of cells. However, its overproduction contributes to oxidative stress, which damages cells. Owing to their biocompatibility, peptidyl complexes are an attractive option for clinical applications to regulate H_2_O_2_ by enzyme mimics. We report here the synthesis and characterization of four new peptidyl di-copper complexes bearing two coordinating sequences. Characterization of the complexes showed that, depending on the linker used between the two coordinating sequences, their catalytic activity for H_2_O_2_ dismutation, their thermodynamic stability and their resistance to H_2_O_2_ degradation are very different, with (CATm2)Cu_2_ being the most promising catalyst.

## 1. Introduction

Reactive oxygen species (ROS) such as superoxide anion O_2_^•−^ or hydrogen peroxide H_2_O_2_ are by-products of the respiratory chain in aerobic organisms. Though essential for fundamental cellular mechanisms such as signalization, an excess of these species leads to oxidative stress and may cause damage to the cells [1,2,3,4,5]. Thus, their intracellular level is tightly controlled by antioxidant defenses, which include metalloenzymes such as superoxide dismutase (SOD) and catalase (CAT). SODs catalyze the dismutation of O_2_^•−^ through a one-electron exchange process, forming O_2_ and H_2_O_2_ [6,7,8]. H_2_O_2_ concentration is regulated by CAT or glutathione peroxidase [9,10]. H_2_O_2_ is better tolerated in cells than superoxide (nM to low µM, whereas steady superoxide concentration is in the pM range) [11] and HO^•^, which is the most reactive of the three, with a cellular half-life in the ns range [12]. However, H_2_O_2_ can be easily transformed into the most damaging HO^•^ by reacting with intracellular copper (I) complexes or iron through the Fenton reaction. Therefore, the development of CAT mimics that could complement CAT when these enzymes are overwhelmed may contribute to oxidative stress management. While SOD mimics have been extensively studied as potential catalytic therapeutic molecules [13,14,15,16], the exploration of CAT mimics is still in its infancy [17,18,19]. On the one hand, some mononuclear manganese mimics have been tested in cellular models, but they lacked stability in such intricate environments [17]. On the other hand, iron or manganese porphyrins, which are much more stable, have shown beneficial effects on H_2_O_2_-mediated loss of viability in catalase/peroxidase-deficient *E. coli* strains [18]. Finally, a di-copper peptidyl catalase mimic has shown the ability to reduce H_2_O_2_ concentration in HeLa Hyper cells [19]. There are two types of catalases: a (monometallic) heme CAT [20] and a dinuclear manganese CAT [21], present in bacteria such as *Lactobacillus plantarum* [22]. Unlike O_2_^•−^, H_2_O_2_ dismutation by CAT requires two electrons, following the half-reactions:H_2_O_2_ → O_2_ + 2H^+^ + 2e^−^ and H_2_O_2_ + 2H^+^ + 2e^−^ → 2H_2_Owith an overall reaction: 2H_2_O_2_ → O_2_ + 2H_2_O(1)

Therefore, CAT mimics need to be bi-electronic as well. So far, the majority of bioinspired mimics have been mononuclear Fe or Mn porphyrinic complexes [18,23], with the ligand porphyrin being easily oxidizable and participating in the bi-electronic exchange, and di-nuclear non-heme Mn complexes [17,24]. A few examples of nanoparticles and metal–organic framework (MOF) nanoparticles have also been identified as CAT mimics [25,26], as well as a few copper complexes [27,28,29,30,31,32]. However, their catalytic activity was often reported in organic solvents or at non-physiological pH because of their low solubility in aqueous pH 7.5 conditions or the need for an external base. We recently described the first peptidyl di-copper complex with catalase-like activity that was promising in aqueous buffer and in cells [19]. However, its first-order rate for H_2_O_2_ dismutation, *k_cat_*, was still two orders of magnitude smaller than that of most active reported non-peptidyl mimics [18]. In addition, we showed that the complex was rapidly degraded in the presence of H_2_O_2_. Herein, we report a second generation of rationally designed copper peptidyl complexes with enhanced stability and a better affinity for copper, especially on the lowest-affinity binding site. Herein, we describe four new complexes. There is a subtle balance between complexes’ thermodynamic stability, their catalytic activity and their ability to resist H_2_O_2_ degradation. Indeed, a complex with high intrinsic catalytic activity will not be able to react with H_2_O_2_ molecules if it is degraded too quickly. On the other hand, if it shows great thermodynamic stability with Cu(II), which prefers square planar geometries, it may not be able to perform electron transfer or to accommodate Cu(I), which prefers tetrahedral geometry, and/or bind to H_2_O_2_ molecules. In such situations, its catalase-like activity will be low. The four new complexes displayed catalase-like activity, and, very interestingly, some of them showed improved features, such as higher resistance to H_2_O_2_ degradation and greater thermodynamic stability in comparison with the previously described peptidyl copper complex [19].

## 2. Results and Discussion

### 2.1. Design and Synthesis

The screening of a combinatorial library of peptidyl copper(II) complexes combined with a catalase activity-based assay enabled the identification of the first peptidyl di-copper complex with promising CAT activity [19]. This first peptide, called CATm1, is shown in Table 1. Its in-depth study confirmed the desired stoichiometry of one peptide for two copper(II) ions to enable two-electron transfer. EPR characterization of the resulting complex suggested a square planar geometry with an N4 or N3O ligand coordination site around both copper atoms, and fluorescence titration revealed two coordination sites with ^app^K_d1_ = 2.8 ± 0.6 × 10^−6^ and ^app^K_d2_ = 8.5 ± 4.9 × 10^−6^ for the first and second sites, respectively. Moreover, despite promising catalytic activity, the first-order rate of H_2_O_2_ dismutation, *k*_cat_, was still two orders of magnitude smaller than that of most manganese non-peptidyl mimics [18]. In addition, the complex undergoes fast degradation in the presence of H_2_O_2_. Therefore, these results invited the improvement of the overall stability of the complex, especially on the second coordination site. 

To do so, a second generation of peptides was rationally designed with two repetitions of the supposed first binding site, PHYKH, which has a higher affinity for copper (II), without (CATm2) or with a spacer (CATm3, CATm4 and CATm5). As spacers, one (CATm3) or two glycine residues (CATm4 and CATm5) were introduced (see Table 1). The introduction of the glycine linker, leading to a glycine–proline (GP) motif (in CATm3 and CATm4), may favor a turn conformation within the peptide chain [33]. Consequently, in CATm5, the proline residue was removed to possibly evaluate the influence of this turn conformation on the resulting copper complex properties. 

The peptides were synthesized by solid-phase peptide synthesis (SPPS) on rink amide resin using a Fmoc strategy. The coupling steps were performed using *N*,*N*′-diisopropylcarbodiimide (DIC) and 1-hydroxybenzotriazole (HOBt) as coupling agents. Fmoc deprotection was achieved using a solution of piperidine in DMF (20:80 *v*:*v*). Each step was monitored using a colorimetric assay (the Kaiser test). After acetylation of the N-terminus, the peptides were cleaved using a TFA solution containing H_2_O and triisopropylsilane (95:2.5:2.5 *v:v:v*). The peptides were precipitated in cold diethyl ether, washed three times with neat diethyl ether and purified by reverse-phase HPLC. The pure peptides were identified by MALDI-TOF mass spectrometry (Appendix A).

### 2.2. Complexes’ Thermodynamic Stability

One of the main challenges in improving catalyst efficiency is to gain thermodynamic stability of the complexes and, in particular, to enhance the second coordination site’s affinity for copper. In order to gain insights into the stability of the complexes, titration was performed by fluorescence spectroscopy since the four new peptides contain two fluorescent tyrosine residues (Y), which can be excited at 275 nm and emit at 303 nm. The fluorescence of tyrosine was quenched when adding copper, possibly because it is one of the Cu(II) ligands. For each peptide, titrations were performed at least twice in MOPS buffer (50 mM, pH 7.5) in order to (i) confirm the 1:2 peptide:Cu(II) stoichiometry of the complex, as with CATm1, and (ii) determine the association constants of the two coordination sites. Titrations performed at high peptide concentration (ca. 72 µM) showed unambiguously that each peptide is able to coordinate two copper metal ions (Appendix A). The curves obtained during titrations at lower concentrations (10–30 µM) could be fitted with a 1-to-2 peptide:Cu model using HypSpec software (Appendix A). The values are reported in Appendix A and presented in Figure 1 for comparison. 

The association constant for the first binding site underwent an increase of one to two orders of magnitude, except for CATm3, which decreased. The highest association constants were obtained with CATm2 and CATm5. On the other hand, the association constant of the second binding site slightly decreased with CATm3 and CATm4. As for the first binding site, the peptides CATm2 and CATm5 seemed to lead to complexes with a more stable second site. CATm2 has no linker between the two repeated sequences (PHYKH), and CATm5 has two glycines and no proline. Consequently, none of these peptides contains the GP sequence, which seems deleterious to achieving a stable second coordination site. The conformation of the peptides, as non-metalated ligand and copper(II) complexes, were further investigated by circular dichroism (CD). None of the ligands adopted a specific conformation (Appendix A), and all complexes exhibited a similar weak CD signature with two maxima at 230 and 260 nm, revealing a low degree of structuration upon binding. When the spectra were normalized according to the number of residues within each peptide, the spectrum of the (CATm1)Cu_2_ complex showed the most intense bands (Figure 2). This suggests a higher content of a defined conformation for this complex in comparison with the others. However, since the higher intensity of these bands does not correlate with higher affinity constants, this conformation may not be the conformation leading to the most stable complexes.

### 2.3. Kinetic Study of Complex Degradation in the Presence of H_2_O_2_

Previous studies with the (CATm1)Cu_2_ complex indicated its rapid degradation in the presence of H_2_O_2_ [19]. Thus, the new complexes were also investigated for their resistance to degradation by H_2_O_2_. It was shown for (CATm1)Cu_2_ that, in a large excess of H_2_O_2_, the complex is transformed into products with a characteristic absorption band at 289 nm. This most likely results from the modification of the phenol ring of the tyrosine. Notably, this is not the case for the non-metalated ligand or Cu(OAc)_2_ [19]. Similar degradation products were obtained with the newly developed complexes, with UV-vis spectra differing only by the intensities of the absorption maxima (Appendix A). We investigated the degradation kinetics of all complexes to better assess their resistance under catalytic conditions. The formation rate of the degradation products can be written as follows:(2)$$v_{0}=k_{cat}{\left[H_{2}O_{2}\right]}_{0}{\left[\mathrm{catalyst}\right]}_{0}$$
or as
(3)$$v_{0}=k_{obs}{\left[\mathrm{catalyst}\right]}_{0}$$
under pseudo-first-order conditions by using a large excess of H_2_O_2_ during the experiments. Measuring the initial rate at different complex concentrations (20, 50, 100 and 200 μM) enabled the determination of the apparent kinetic constant *k_obs_* (Figure 3 and Table 2). In this experiment, a low *k*_obs_ indicates a high resistance to H_2_O_2_ degradation.

The complexes (CATm2)Cu_2_, (CATm3)Cu_2_ and (CATm4)Cu_2_ exhibited a smaller *k*_obs_ than (CATm1)Cu_2_, meaning their degradation is slower. On the other hand, (CATm5)Cu_2_ degradation was faster than (CATm1)Cu_2_ degradation. The overall order of degradation, from the slowest to the fastest, is (CATm3)Cu_2_ < (CATm4)Cu_2_ < (CATm2)Cu_2_ < (CATm1)Cu_2_ < (CATm5)Cu_2_. This does not correlate with the thermodynamic stability of the complexes, suggesting that the degradation of the complexes may not be due to released Cu(I) that could have reacted with H_2_O_2_ to form radicals such as HO^•^. Indeed, the complexes’ degradation may also result from a reaction with HO^•^, which is generated by the complexes themselves, as has been described for amyloid peptide/Cu complexes [34].

### 2.4. Catalytic Activity

The ability of the four complexes to catalyze hydrogen peroxide dismutation and to thus mimic the enzyme CAT was then investigated. A Clark-type electrode can be used to monitor O_2_ formation when H_2_O_2_ is added to the complex solution (see Appendix A for representative experiments). The catalase activity of many complexes mimicking CAT has only been studied in organic solvents or at relatively high pH [29,32,35], except for a few [31,35,36,37]. With peptidyl complexes, aqueous solubility at pH around 7 allowed us to conduct studies in more biologically relevant conditions, namely, in aqueous MOPS buffer (50 mM) at pH 7.5. While the initial concentration of the complex CATmx:Cu(OAc)_2_ 1:2 was held constant at 100 μM, the reaction was studied with variable amounts of H_2_O_2_ (from 2.5 mM to 30 mM). Endogenous MnCAT exhibits Michaelis–Menten catalytic behavior [38]. The initial rate of O_2_ formation satisfies the Michaelis–Menten equation: (4)$$v_{0}=\frac{v_{\max}{\left[H_{2}O_{2}\right]}_{0}}{K_{M}+{\left[H_{2}O_{2}\right]}_{0}}$$
where *v*_0_ is the initial rate, *v*_max_ is the maximum rate for a given catalyst concentration, *K*_M_ is the Michaelis–Menten constant and is a measure of the catalyst affinity for H_2_O_2_ (the lower the *K*_M_, the higher the affinity), and [H_2_O_2_]_0_ is the initial substrate concentration. We can also write:(5)$$v_{\max}=k_{cat}{\left[\mathrm{Catalyst}\right]}_{0}$$
with *k*_cat_ being the catalytic rate constant for H_2_O_2_ dismutation. These different catalytic parameters can be easily determined using the Lineweaver–Burk method involving the reciprocal of Equation (5):(6)$$\frac{1}{v_{0}}=\left(\frac{K_{M}}{v_{\max}}\right)\left(\frac{1}{{\left[H_{2}O_{2}\right]}_{0}}\right)+\frac{1}{v_{\max}}$$

When 1/*v*_0_ is plotted against 1/[H_2_O_2_]_0_, the slope of the obtained line gives *K*_M_*/v*_max_, and the intercept with the x-axis (abscissa) gives −1/*K*_M_. *k*_cat_ is then calculated according to Equation (5) (Figure 4a). Experiments were repeated at least twice to ensure the consistency of the results, which are summarized in Table 3.

Dismutation transforms 2 moles of H_2_O_2_ into 1 mole of O_2_ (Equation (1)). Hence, the catalytic rate constant *k*_cat_ for H_2_O_2_ dismutation corresponds to 2**k*_cat_ for O_2_ formation. All kinetic parameters reported hereafter are according to H_2_O_2_, even if it is O_2_ evolution that was monitored. The results are reported in Table 3. The catalytic rates *k*_cat_ of the four new complexes are in the same order of magnitude as (CATm1)Cu_2_. (CATm3)Cu_2_ showed the best catalytic rate constant (1.3 × 10^−1^ s^−1^) but also the weakest affinity for H_2_O_2_ (*K*_M_ = 52 mM). In order to take into account these two features, the ratio *k*_cat_/*K*_M_, which better reflects catalytic efficiency, was calculated. CATm2 is the peptide leading to the most efficient complex since its *k*_cat_/*K*_M_ ratio (3.9 M^−1^ s^−1^) is the highest, followed by CATm5 (3.2 M^−1^ s^−1^). Interestingly, these data are consistent with the thermodynamic constants measured by fluorescence since (CATm2)Cu_2_ and (CATm5)Cu_2_ are the most thermodynamically stable complexes (Figure 1). The apparent second-order constant (*k*_cat_/*K*_M_) for the 1:2 complexes is higher than the values reported for other copper(II) complexes, except for CuL_2_ (Table 3).

The overall efficiency of a CAT mimic must be evaluated while taking into account its reactivity towards H_2_O_2_ and its resistance to degradation. To that end, the turnover numbers (TONs) were calculated using the previous kinetic data (Figure 4b). The TON is the number of moles of converted substrate (H_2_O_2_) per mole of catalyst and gives an estimation of the longevity of a catalyst. For all complexes, the TON increased when H_2_O_2_ concentration increased to 7.5 mM, with (CATm2)Cu_2_ always possessing the highest TON. This is consistent with its higher *k*_cat_/*K*_M_ ratios. At 10 mM, the four catalysts (CATm1)Cu_2_, (CATm2)Cu_2_, (CATm4)Cu_2_ and (CATm5)Cu_2_ exhibited similar TON values, whereas the least efficient catalyst (CATm3)Cu_2_ showed weaker TON values. At high H_2_O_2_ concentrations (25 and 30 mM), the TON value of (CATm5)Cu_2_, the most fragile complex (according to the previous experiment; see Figure 3), dropped, whereas the TON value of (CATm3)Cu_2_, which is the more resistant complex to H_2_O_2_ degradation, kept increasing. This clearly shows that at these concentrations, the degradation of the catalyst becomes a key parameter. The TON values did not increase when the experiments were performed in the presence of D-mannitol, a HO^•^ quencher (up to 1 mM, data not shown), suggesting that the complexes were not degraded because of HO^•^ formation.

## 3. Conclusions

Ultimately, the overall efficacy of a complex as a CAT mimic depends on a fine balance between several parameters, including its reactivity towards H_2_O_2_; its own stability and affinity for its metal, determining its survival in biological conditions; and its resistance to H_2_O_2_ degradation, which remains a strong oxidant. In order to take into account all of these features, radar plots were used for comparison purposes (Figure 5). We chose to report the three main parameters that play an important role in the catalysis (thermodynamic stability, dismutation kinetics and resistance to degradation). The axes were chosen so that an increase was associated with an improvement: hence, we chose the overall stability of the complexes (log (Ka1*Ka2)), the kinetics of the dismutation *k*_cat_/*K*_M_ and the inverse of the degradation rate. In such radar plots, the higher the surface, the better the catalyst.

Using such representations, it clearly appears that the area in the case of (CATm2)Cu_2_ is the widest, indicating that this complex is the most promising catalyst. Interestingly, the CATm2 sequence is the shortest of the four new sequences investigated, suggesting that compact structures may be more suitable for developing efficient catalysts. These encouraging results call for further investigations in cellular models of oxidative stress, as this catalyst already possesses interesting properties. Overall, the rational approach adopted in this work led to a noticeable but weak improvement of the catalyst. This underlines the fact that designing a peptidyl sequence that is able to accommodate two metal centers and lead to complexes with the expected properties is not straightforward. Combining a combinatorial approach with activity-based screening may be a more valuable strategy in this case. Works in this direction are currently in progress.

## 4. Experimental Section

**Peptide synthesis.** Peptide synthesis was conducted by SPPS using Fmoc-Rink Amide MBHA (resin with a loading capacity of 0.53 mmol/g). Solid-phase peptide synthesis was performed manually with standard Fmoc-protected amino acids. Resin beads at a concentration of 0.200 mmol were first swelled in dichloromethane (DCM). Standard peptide coupling procedures were used for all amino acid couplings: the amino acid (3 equiv./equiv. resin) with *N,N*’-diisopropylcarbodiimide (DIC; 3 equiv./equiv. resin) and *N*-hydroxybenotriazole (HOBt; 3 equiv./equiv. resin) in dimethylformamide (DMF, 4 mL) were mixed with the resin for 1 h at room temperature under agitation. After each amino acid coupling, solvents and soluble reagents were removed under vacuum, and the resin was washed five times with DMF. Completion of the reaction was monitored by the Kaiser test, which indicates the presence of free amine by the deep blue coloration of the bead. Standard deprotection conditions of the Fmoc group were employed (20% piperidine in N-methyl-2-pyrrolidone (NMP) for 1 min under agitation and again for 15 min under agitation at room temperature), followed by washings with DMF. Final acetylation was performed with a solution of acetic anhydride in dichloromethane DCM (4 mL, 10/90, *v:v*) for 1h at room temperature. Solvents and soluble reagents were removed by filtration. The resin was successively washed with DCM (5 × 4 mL) and then methanol (3 × 4 mL) and dried under vacuum for 1h. Simultaneous lateral chain deprotection and bead cleavage were performed using a solution of TFA/H_2_O/Trisopropylsilane (95%/2.5%/2.5%; 4 mL) (TFA, trifluoracetic acid) for 2h at room temperature. The samples became red/orange. The solution was collected by filtration in a 50 mL round-bottom flask, and beads were washed three times with neat TFA (3 × 3 mL). The TFA solutions were combined, and TFA was removed under reduced pressure. The crude solid was precipitated in cold diethyl ether and recovered by centrifugation (7000 rpm, 4 min). The solid was washed two more times with Et_2_O. The peptide was dissolved in 20 mL of deionized H_2_O and freeze-dried. The peptide was purified by reverse-phase HPLC using a linear gradient from 5 to 30% acetonitrile in a water bath containing 0.1% TFA for 30 min. High purity (>95%) was confirmed by analytical HPLC, and the expected mass was found by MALDI-TOF mass spectrometry. The list of the synthesized peptides is provided in Appendix A.

**Peptide stock solution preparation.** The concentration of purified and lyophilized peptide dissolved in milliQ water was determined by measuring the absorbance of a diluted solution (5 µL in 995 µL of milliQ water) at 280 nm and by using the extinction coefficient of tyrosine (1280 cm^−1^M^−1^).

**Association constant measurement by fluorescence spectrometry.** Cu(OAc)_2_·H_2_O was titrated into CATmx (x = 1–5) peptide solution in MOPS buffer (50 mM, pH 7.5) at 25 °C. After each addition of Cu(II), the emission spectrum (average of 2 accumulations) upon excitation at 275 nm was recorded between 280 and 400 nm (slit_exc_ = slit_em_ = 5 nm; scan rate = 200 nm/min). Before recording each spectrum, it was ensured that thermodynamic equilibrium was reached (stable fluorescence intensity). The intensity at the maximum of tyrosine emission (303 nm) was used to generate titration curves, which were then fitted using HypSpec software. HypSpec determines cumulative (β) binding constants for the first (β1) and second (β2) Cu(II) bindings to the peptide. From these cumulative binding constants, stepwise binding constants for the first (K_1_) and second (K_2_) Cu(II) bound to the peptide were determined. The provided K_d_ values are the average of two to three independent titrations ± standard error of the mean (SEM).

**Kinetic study of complex degradation by UV–visible spectroscopy.** Complex (CATmx:Cu 1:2) degradation kinetics was monitored for each complex (x = 1–5) at 4 different concentrations (20, 50, 100 and 200 µM) in MOPS buffer (50 mM, pH 7.5). The absorbance was recorded at 289 nm over time until a plateau was reached. The addition of H_2_O_2_ in excess (5 mM) to the cuvette corresponds to the beginning of the experiment (t0). The initial rates of degradation product formation *v*_0_ (taking into account their molar extinction coefficient (ε); see SI) were measured from the slope of the linear fit (from 0 to 30 s) for the different complex concentrations. For each complex concentration, measurements were performed twice, and the values used to plot Figure 3 are average ± standard error of the mean (SEM). Then, *k*_obs_ corresponds to the slope of the linear fit of *v*_0_ = f([complex]).

**Catalytic activity.** All measurements were carried out in MOPS buffer (50 mM, pH 7.5) at 25 °C in a micro-cell sealed with a rubber septum to avoid the introduction of O_2_ from the air. The complex solution CATmx:Cu(OAc)_2_ 1:2 (x = 1–5) at 100 µM in MOPS was introduced into the micro-cell and sealed before being bubbled with dinitrogen gas to remove dissolved dioxygen. H_2_O_2_ solution was then injected through the septum into the stirred complex solution. Reaction rates were determined by measuring the O_2_ concentration evolution over time. The initial rates *v*_0_ were determined from the slope of the linear fit (from 0 to 20 s) of dioxygen formation at several initial H_2_O_2_ concentrations (2.5, 5, 7.5, 10, 15, 20, 25 and 30 mM). The O_2_ formation reaction from H_2_O_2_ exhibited Michaelis–Menten catalytic behavior, as is the case for the natural enzyme. Michaelis–Menten constants *K*_m_, V_max_ and *k*_cat_ were determined using the Lineweaver–Burk double reciprocal plot: 1/*v*_0_= f(1/[H_2_O_2_]), in which the x-intercept corresponds to −1/*K*_m_, the slope corresponds to *K*_m_/V_max_ and V_max_ = *k*_cat O2_[catalyst]_0_. Since, during dismutation, two H_2_O_2_ molecules lead to the formation of a single O_2_ molecule, *k*_cat-disappearance H2O2_ = 2 × *k*_cat apparition O2_. The TON was calculated as the maximum number of H_2_O_2_ moles consumed per mole of catalyst, TON = 2[O_2_]_max obs_/[catalyst]_0_.

## Figures and Tables

**Figure 1 molecules-27-05476-f001:**
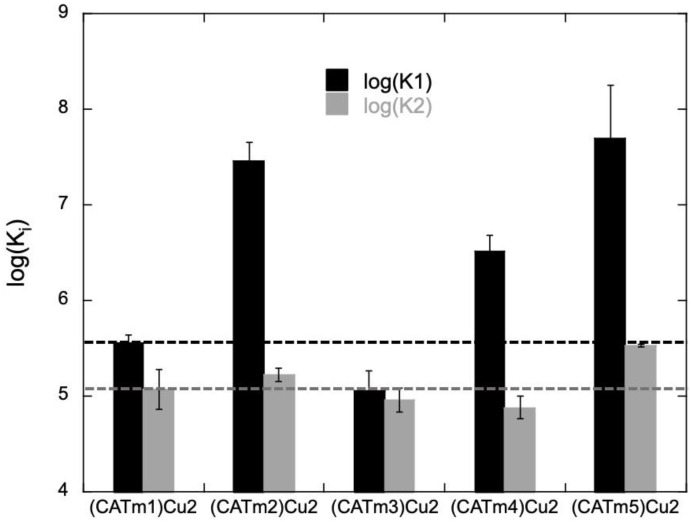
Comparison of log(^app^K_1_) and log(^app^K_2_) of the 1:2 peptide:Cu complexes. The apparent association constants of the two binding sites for each complex were measured using fluorescence spectroscopy at 25 °C in MOPS buffer (50 mM, pH 7.5) using HypSpec software. HypSpec determined cumulative (β) binding constants for the first (β1) and second (β2) Cu(II) bindings to the peptide. From these cumulative binding constants, the stepwise binding constants for the first (K1) and second (K2) Cu(II) bound to the peptide were determined. The provided ^app^K values are the average of two to three independent titrations ± standard error of the mean (SEM). Excitation was set at 275 nm, and spectra were recorded from 280 to 400 nm (see Experimental Section for more details).

**Figure 2 molecules-27-05476-f002:**
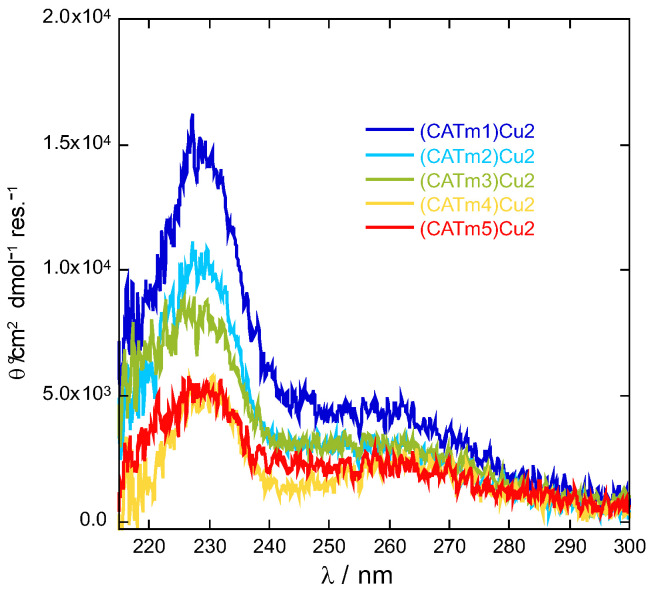
Circular dichroism spectra normalized by the number of residues within each peptide of CATmx:Cu 1:2 mixtures ([CATmx] = 133 µM, x = 1–5). Spectra were recorded at 20 °C in MOPS buffer (50 mM, pH 7.5).

**Figure 3 molecules-27-05476-f003:**
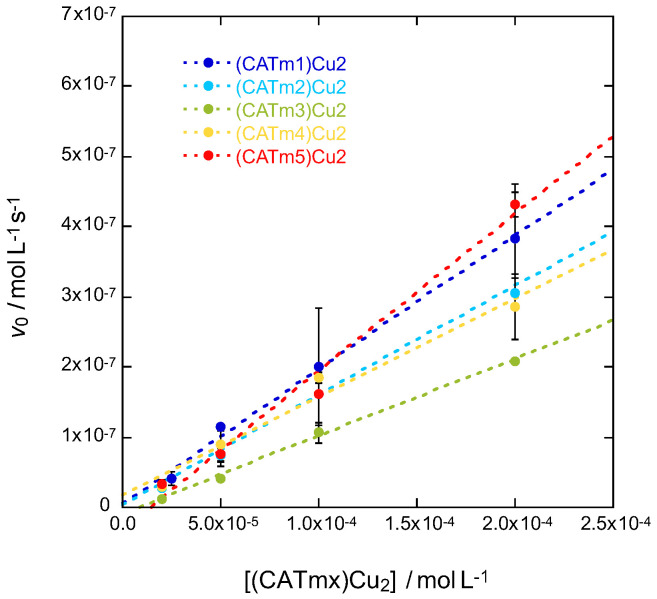
Determination by UV-vis spectrometry at 289 nm of the apparent kinetic constants of degradation product formation for complexes (CATmx)Cu_2_ (x = 1 to 5) in MOPS (50 mM, pH 7.5) with an excess of H_2_O_2_ (5 mM).

**Figure 4 molecules-27-05476-f004:**
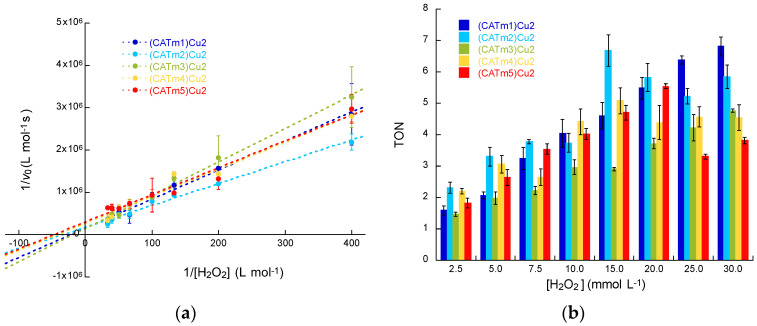
Catalytic behavior of the five CATmx:Cu 1:2 complexes in MOPS (50 mM, pH 7.5) at 25 °C. Experiments were performed with [(CATmx)Cu_2_] = 100 µM (x = 1–5). (**a**) Lineweaver–Burk plot (double reciprocal of the Michaelis–Menten equation), allowing the determination of enzyme kinetic parameters. (**b**) Turnover numbers (TONs) of dismutated H_2_O_2_ measured at various [H_2_O_2_] concentrations. Data are given as the average of at least 2 experiments ± standard error of the mean (SEM). Notably, kinetics values for [(CATm1)Cu_2_ are slightly different from previously reported data [19] since, in this work, kinetics were measured in a larger [H_2_O_2_] concentration range, which led to a different slope in Figure 4a.

**Figure 5 molecules-27-05476-f005:**
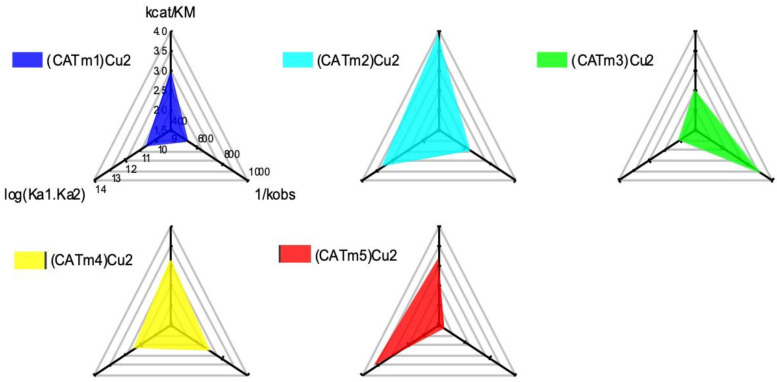
Comparison of catalyst performance using radar representation. *k*_cat_/*k*_M_ are parameters related to the kinetics of H_2_O_2_ dismutation, K_a1_ and K_a2_ are the association constants of the two binding sites, and *k*_obs_ is the rate of degradation of the catalyst in the presence of an excess of H_2_O_2_.

**Table 1 molecules-27-05476-t001:** Sequence of the peptidyl ligands studied. In bold are highlighted residues that may favor a turn conformation.

Name	Peptide Sequence
CATm1	Ac(PHYKH)RLH-NH_2_
CATm2	Ac(PHYKH)(PHYKH)-NH_2_
CATm3	Ac(PHYKH)**G**(**P**HYKH)-NH_2_
CATm4	Ac(PHYKH)G**G**(**P**HYKH)-NH_2_
CATm5	Ac(PHYKH)GGHYKH-NH_2_

**Table 2 molecules-27-05476-t002:** Rates of catalyst degradation under pseudo-first-order conditions ([H_2_O_2_] = 5 mM) at 25 °C in MOPS buffer (50 mM, pH 7.5) obtained from the slopes of the lines in Figure 3.

	*k*_obs_ (s^−1^)
(CATm1)Cu_2_ ^a^	1.90 × 10^−3^
(CATm2)Cu_2_	1.56 × 10^−3^
(CATm3)Cu_2_	1.10 × 10^−3^
(CATm4)Cu_2_	1.41 × 10^−3^
(CATm5)Cu_2_	2.24 × 10^−3^

^a^ The value is different from the value reported in Reference [19] because the extinction coefficient was measured differently (see Appendix A for more details about extinction coefficient determination and Appendix A).

**Table 3 molecules-27-05476-t003:** Parameters describing the catalysis of H_2_O_2_ dismutation.

	*K*_M_ (M)	*k*_cat_ (s^−1^)	*k_c_*_at_/*K*_M_ (M^−1^·s^−1^)	References
(CATm1)Cu_2_ ^[a]^	4.8 × 10^−2^	1.4 × 10^−1^	2.9	This work, [19]
(CATm2)Cu_2_ ^[a]^	2.9 × 10^−2^	1.1 × 10^−1^	3.9	This work
(CATm3)Cu_2_ ^[a]^	5.2 × 10^−2^	1.3 × 10^−1^	2.5	This work
(CATm4)Cu_2_ ^[a]^	2.4 × 10^−2^	0.8 × 10^−1^	3.1	This work
(CATm5)Cu_2_ ^[a]^	2.1 × 10^−2^	0.7 × 10^−1^	3.2	This work
Cu(N-baa)_2_(phen) ^[b]^	5.2 × 10^−2^	6.6 × 10^−2^	1.3	[29]
[Cu(HL1)]^2+ [c]^	1.7 × 10^1^	1.5 × 10^−3^	8.9 × 10^−5^	[31]
Cu_2_(pxdiprbtacn)Cl_4_ ^[d]^	1.5	1.24	0.8	[36]
CuL_2_ ^[e]^	4.2 × 10^−2^	3.6 × 10^−1^	8.25	[37]
[Cu(apzpn)]^2+ [f]^			1.10	[35]
[Cu(py2pn)]^2+ [g]^		0.8 × 10^−4^		[32]
*Catalase*	8.3 × 10^−2^	2.6 × 10^5^	3.1 × 10^6^	[38]

^[a]^ *k*_cat_ is the first-order rate of H_2_O_2_ dismutation, and *K*_M_ is the Michaelis–Menten constant. In this work, they were measured with the CATm1:Cu^2+^ 1:2 mixture at 100 µM. Reactions were performed in MOPS buffer (50 mM, pH 7.5) at 25 °C. ^[b]^ *N*-baaH: N-benzoylanthranilic acid; phen: 1,10-phenanthroline. The reactions were performed in DMF at 20 °C. ^[c]^ HL^1^: 1,3-bis[(2-aminoethyl)amino]-2-propanol. The reactions were performed at 25 °C in TRIS buffer, pH 7. ^[d]^ pxdiprbtacn: 1,4-Bis(4,7-diisopropyl-1,4,7-triazacyclonon-1-ylmethyl)benzene. The reactions were performed in phosphate buffer (0.01 M, pH = 7.4) at 25 °C. ^[e]^ L^2^: 2-{[(3-chloro-2-hydroxy-propyl)-pyridin-2-ylmethyl-amino]-methyl}-phenol. The reactions were performed in phosphate buffer solution at pH 7.8. ^[f]^ apzpn: N,N’-bis(2-acetylpyrazyl)methylene-1,3-diaminopropane). The reactions were performed at 30.0 ± 0.1 °C in borate buffer (0.10 M, pH 8). ^[g]^ py_2_pn: *N,N′*-Bis(2-pyridinylmethylen)propane-1,3-diamine). The reactions were performed at 25 °C in DMF solution of the complex containing 100 mM of Et_3_N.

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
