# Peer review of "Improvement of Peptidyl Copper Complexes Mimicking Catalase: A Subtle Balance between Thermodynamic Stability and Resistance towards H2O2 Degradation"

_molecules, 2022, doi:10.3390/molecules27175476_

Round 1
Reviewer 1 Report
The article describes the synthesis of four new peptidyl copper complexes which mimics the activity of the Catalase enzyme catalyzing the decomposition of hydrogen peroxide to water and oxygen. The structures of this peptidyl copper complexes are based on a previous investigation of the group and are showing mainly better activities than the previously used complex (CATm1)Cu2) which was also used for comparison in this article. The new ligands are variations of the ligand used in the (CATm1)Cu2)-complex. The peptide sequence was doubled and by introducing spacers of one and two glycine moieties and one glycine moiety without proline, they could get four new ligands.
The introduction and the theoretical backround shows clearly the motivation for this investigation supported by sufficient current bibliography. (Citation 18: The list of Authors is not complete, should be completed.)
The thermodynamic stability of the complexes was shown by association measurements by fluorescence spectroscopy titrating peptide solutions with copper acetate and measuring emissions spectrum after each adding of copper acetate. In conclusion they formed principally complexes of one ligand with one copper ion. The complex (CATm5)Cu2 results to be the most stable at the first and second binding side followed by (CATm2)Cu2 complex due probably to the absence of a GP sequence at the peptidyl ligand. The type of experiment used are very good to determine this relationship and is up to date. Structures of these complexes were not reported.
I think there is only one error In line 117. I think it should mean: “…., which seems (deleterious) beneficial to get a more stable second…”.
The conducted kinetic studies take into account that H2O2 transforms the complexes, and only the complexes, in an unspecified product which has a characteristic absorption peak at 289 nm. The results were interpreted as a pseudo first order reaction by using an excess concentration of H2O2. The overall order of degradation of the complexes do not correlate with the thermodynamic stability of the complexes.
The catalytic activities of the complexes synthesized in this work were measured under more biological conditions which was never done before because of a low solubility of previously complexes described in literature.
One little error in the table 3: KM (M), the parenthesis should be correct.
The work leads to the conclusion that there are three mayor parameters playing an important role for catalytic behavior: thermodynamic stability, dismutation kinetics and resistance toward degradation of the complexes.
The experimental section shows detailed information of the conducted experiments.
The article shows an advance in the development for new and advanced mimics of the Catalase enzyme with better stability, catalytic activity and dismutation kinetics for this purpose. I recommend the publication with some minor errors which I mentioned here and they should be attended.
Author Response
The article describes the synthesis of four new peptidyl copper complexes which mimics the activity of the Catalase enzyme catalyzing the decomposition of hydrogen peroxide to water and oxygen. The structures of this peptidyl copper complexes are based on a previous investigation of the group and are showing mainly better activities than the previously used complex (CATm1)Cu2) which was also used for comparison in this article. The new ligands are variations of the ligand used in the (CATm1)Cu2)-complex. The peptide sequence was doubled and by introducing spacers of one and two glycine moieties and one glycine moiety without proline, they could get four new ligands.
The introduction and the theoretical backround shows clearly the motivation for this investigation supported by sufficient current bibliography. (Citation 18: The list of Authors is not complete, should be completed.)
We have added the lacking names.
The thermodynamic stability of the complexes was shown by association measurements by fluorescence spectroscopy titrating peptide solutions with copper acetate and measuring emissions spectrum after each adding of copper acetate. In conclusion they formed principally complexes of one ligand with one copper ion. The complex (CATm5)Cu2 results to be the most stable at the first and second binding side followed by (CATm2)Cu2 complex due probably to the absence of a GP sequence at the peptidyl ligand. The type of experiment used are very good to determine this relationship and is up to date. Structures of these complexes were not reported.
We thank the referee for these nice remarks. Elucidation of the structures would be very interesting but is not in the scope of this article.
I think there is only one error In line 117. I think it should mean: “…., which seems (deleterious) beneficial to get a more stable second…”.
It seems that there is a confusion from the reviewer. Both CATm3 and CATm4 possess a GP sequence and exhibit a lower K for the second site than CATm1, consequently, it seems that this GP sequence is deleterious to reach high association constants.
The conducted kinetic studies take into account that H2O2 transforms the complexes, and only the complexes, in an unspecified product which has a characteristic absorption peak at 289 nm. The results were interpreted as a pseudo first order reaction by using an excess concentration of H2O2. The overall order of degradation of the complexes do not correlate with the thermodynamic stability of the complexes.
The catalytic activities of the complexes synthesized in this work were measured under more biological conditions which was never done before because of a low solubility of previously complexes described in literature.
One little error in the table 3: KM (M), the parenthesis should be correct.
The bracket is correct in the manuscript.
The work leads to the conclusion that there are three mayor parameters playing an important role for catalytic behavior: thermodynamic stability, dismutation kinetics and resistance toward degradation of the complexes.
The experimental section shows detailed information of the conducted experiments.
The article shows an advance in the development for new and advanced mimics of the Catalase enzyme with better stability, catalytic activity and dismutation kinetics for this purpose. I recommend the publication with some minor errors which I mentioned here and they should be attended.
We thank the referee for the encouraging remarks concerning the manuscript
Reviewer 2 Report
This manuscript presents the preparation of second-generation copper peptidyl complexes with enhanced stability, a better affinity for copper, and a higher resistance toward H2O2 degradation. The results are interesting, and the analyses are reasonable. I recommend the acceptance of the manuscript after some minor revisions.
1) A schematic diagram should be given to illustrate the preparation of the complexes.
2) Page 2, line 38 – “the exploration of CAT mimics is still at its infancy [17–19].” Please briefly describe what type of CAT mimics were investigated in these references.
3) In general, the grammar used in this paper can be improved. I suggest that some of the professional English language authors, or an editor, take a more active role in the re-writing and editing of the manuscript.
4) The conclusion section should be improved, and it is suggested to introduce the limitations of this research briefly.
Author Response
This manuscript presents the preparation of second-generation copper peptidyl complexes with enhanced stability, a better affinity for copper, and a higher resistance toward H2O2 degradation. The results are interesting, and the analyses are reasonable. I recommend the acceptance of the manuscript after some minor revisions.
1) A schematic diagram should be given to illustrate the preparation of the complexes.
Complexes are prepared simply by mixing the respective peptides with copper acetate in a 1:2 ratio in a buffer. We are not sure this requires a specific scheme as it is very common.
2) Page 2, line 38 – “the exploration of CAT mimics is still at its infancy [17–19].” Please briefly describe what type of CAT mimics were investigated in these references.
We have added a brief description of the cited papers content:
One the one hand, some mononuclear manganese mimics have been tested in cellular models but they lacked stability in such intricate environments [17]. On the other hand, iron or manganese porphyrins, which are much more stable, have shown beneficial effects on H2O2-mediated loss of viability in catalase/peroxidase-deficient E. coli strains [18]. Finally, a di-copper peptidyl catalase mimic has shown ability to reduce H2O2 concentration in HeLa Hyper cells [19].
3) In general, the grammar used in this paper can be improved. I suggest that some of the professional English language authors, or an editor, take a more active role in the re-writing and editing of the manuscript.
We went through the manuscript and we have modified some errors that are highlighted in yellow
4) The conclusion section should be improved, and it is suggested to introduce the limitations of this research briefly.
We have modified the conclusion accordingly.
Reviewer 3 Report
The paper by Delsuc et al. represents research focusing on synthesizing di-copper peptidyl complexes, their catalytic activity for H2O2 dismutation, their thermodynamic stability, and their resistance toward H2O2 degradation. This work is a part of the ongoing research of the group. The paper is well written, the results are interesting and seem to be correct, and the research is suitable for publication in Molecules, but needs minor revisions:
1. First of all, there is no supporting information containing a lot of important information. Several pictures, for example, luminescent titration with corresponding saturation graphs suggested to being given in the main text.
2. Authors should explain how they calculate the dissociation constants and give the corresponding dimension (units) for values (mol/l?). During the reading of the manuscript, the calculated dissociation constants seem to be incorrect. For example, K1 looks like an overall constant to the formation of 1:1 complex (mol/l), but K2 looks to be an overall constant to the formation of 1:2 complex (mol^2/l^2). The authors use cumulative and stepwise binding constants, but these terms arise only in the experimental part and mislead. The corresponding equation for such calculations can help avoid this confusion
Author Response
The paper by Delsuc et al. represents research focusing on synthesizing di-copper peptidyl complexes, their catalytic activity for H2O2 dismutation, their thermodynamic stability, and their resistance toward H2O2 degradation. This work is a part of the ongoing research of the group. The paper is well written, the results are interesting and seem to be correct, and the research is suitable for publication in Molecules, but needs minor revisions:
- First of all, there is no supporting information containing a lot of important information. Several pictures, for example, luminescent titration with corresponding saturation graphs suggested to being given in the main text.
We have added all titration curves performed by fluorescence as suggested by the referee. In addition, the great majority of the experimental details are given in the experimental part.
- Authors should explain how they calculate the dissociation constants and give the corresponding dimension (units) for values (mol/l?). During the reading of the manuscript, the calculated dissociation constants seem to be incorrect. For example, K1 looks like an overall constant to the formation of 1:1 complex (mol/l), but K2 looks to be an overall constant to the formation of 1:2 complex (mol^2/l^2). The authors use cumulative and stepwise binding constants, but these terms arise only in the experimental part and mislead. The corresponding equation for such calculations can help avoid this confusion
We have added the details about association constant determination in the legend of Figure 1. K1 and K2 are stepwise binding constants associated to the two following reactions.
Cu + L -> CuL (K1) and CuL + Cu -> Cu2L (K2)
Since they are ratio of activities, association constants (and dissociation constants) are formally dimensionless. They are often reported with unit such as M-1 (or M) but it is not correct that is why the association constants reported in the text are given without dimension.